

# Significance of urine complement proteins in monitoring lupus activity

Jin Zhao[1,*], Jun Jiang[2,*], Yuhua Wang[1], Dan Liu[2], Tao Li[1] and Man Zhang[1,2,3]

[1] Beijing Shijitan Hospital, Capital Medical University, Beijing, China
[2] Peking University Ninth School of Clinical Medicine, Beijing, China
[3] Beijing Key Laboratory of Urinary Cellular Molecular Diagnostics, Beijing, China
[*] These authors contributed equally to this work.

## ABSTRACT

**Objectives**. Complement activation is a critical feature in the development of systemic lupus erythematosus (SLE). Whether there are changes of complement components in the urine of SLE has not been reported. The aim of the study was to evaluate the complement-related proteins in the urine of SLE, verify differentially expressed proteins(DEPs) in the active phase of SLE, further explore their clinical application value.

**Methods**. First, we used bioinformatics and functional enrichment to screen and identify the urine protein profile of SLE patients. Then, analyzed and verified the proteins related to the complement pathway by western-blot and Parallel Reaction Monitoring (PRM) technology. Further evaluated the relationship between urinary DEPs related to complement pathway and disease activity.

**Results**. A total of 14 complement pathway-related proteins were screened for differences in expression between the active group and the stable group, eight of these DEPs were up-regulated and six were down-regulated. These DEPs may play a key role in SLE disease activity. We used PRM technology to verify the eight up-regulated proteins, and found that four of these complement proteins, namely C9, C8A, C4B, and C8G, were significantly increased in active group. Furthermore, these four DEPs were highly correlated with disease activity. In the urine of SLE patients, AUCs of 0.750, 0.840, 0.757 and 0.736 were achieved with C9, C8A, C4B, and C8G, respectively.

**Conclusions**. Complement-related DEPs in urine have a certain correlation with SLE disease activity. Urine C9, C8A, C4B and C8G present promising non-invasive biomarkers for monitoring lupus activity.

# INTRODUCTION

Systemic lupus erythematosus (SLE) is a global chronic autoimmune disease involving multiple organs. With the development of diagnosis and treatment technology, the survival rate of SLE has increased significantly, but it is still an irradicable disease (*Tamirou et al., 2018*). SLE has the characteristics of repeated recurrence and remission. Relapsing is a common clinical feature, a sign of obvious disease activity, and the main cause of organ damage and poor prognosis (*Ocampo-Piraquive et al., 2018*). Autoantibodies, immune

Corresponding author
Man Zhang, zhangman@bjsjth.cn

complexes and complement system in the immune system play an important role in the pathogenesis of SLE (*Kiriakidou & Ching, 2020*; *Fava & Petri, 2019*). Complement is an important effector system of the immune response. On the one hand, complement is the ultimate effector of tissue damage in autoimmune diseases; on the other hand, the lack of certain components of complement would lead to autoimmune diseases (*Weinstein, Alexander & Zack, 2021*).

It has been found that many complement components were closely related to the occurrence and development of SLE. Serum low levels of complement C3, C4 and CH50 as indicators of lupus activity have been included in the Systemic Lupus Erythematosus Disease Activity Index 2000 (SLEDAI-2000) scoring system (*Gladman, Ibanez & Urowitz, 2002*). The relationship between serum C1q and the development of lupus is well known, it is typically involved in the clearance of apoptotic cells, and being a valuable biomarker of lupus activity (*Thielens et al., 2017*; *Sandholm et al., 2019*). The genetic defects of complement C1, C2, or C4 are strongly associated with the development of SLE (*Leffler, Bengtsson & Blom, 2014*). Since most complement and activation products exist in plasma and are finally excreted in urine, these factors may be used as non-invasive biomarkers for monitoring disease activity. In our early study, we used label-free technology to analyze the urine proteins between the active and stable group of SLE, and discovered many differentially expressed proteins (DEPs) (*Jiang et al., 2022*). The changes of complement-related proteins in the active phase of SLE aroused our interest.

Therefore, in order to analyze the expression of complement-related proteins in the urine of SLE, we evaluated the urine protein expression profile during the active period of SLE. Subsequently, we verified the changes of these complement proteins in the urine of lupus by qualitative western-blot and quantitative proteomics technology, further explored the role of different complement components in lupus activity to provide a basis for discovering urinary biomarkers of lupus activity.

## MATERIAL AND METHODS

### Patients and sample collection

First of all, this study protocol was approved by the ethics committee of Beijing Shijitan Hospital, Capital Medical University (sjtkyll-lx-2021(55)). The participants all gave written informed consent of each subject, which was in accordance with the provisions of the Helsinki Declaration. All subjects were adult females. Study participants included 24 patients with diagnosed systemic lupus erythematosus (SLE group) and 12 controls that did not have SLE (NC) with routine physical examination. Participants were recruited from Beijing Shijitan Hospital from September 2020 until September 2021. SLE patients were diagnosed through the use of 1997 American College of Rheumatology classification criteria for systemic lupus erythematosus. According to the SLEDAI-2K scoring system, SLEDAI-2K ≥5 were classified as the active group (SLE-A), and SLEDAI-2K<5 were classified as the stable group (SLE-S). Exclusion criteria included acute or chronic infections, tumors and other autoimmune diseases, and any diagnoses of other severe liver and renal disease. The morning midstream urine samples were collected into sterile polypropylene tubes.

Immediately after collection, urine samples were centrifuged at 400 ×g for 5 min to remove cell debris and casts and finally supernatants were divided in aliquots and frozen at −80 °C.

## Proteomic analysis and bioinformatics analysis

Base on SDS Polyacrylamide Gel Electrophoresis (SDS-PAGE) combined with nano-upgraded reversed-phase liquid chromatography-tandem mass spectrometry (nanoRPLC-MS/MS) method, we detected and analyzed the urinary proteins of SLE patients and controls without SLE using Thermo Q-Exactive mass spectrometer. MaxQuant software was used to process the original mass spectrometry files, and the significant difference in protein quantification between SLE-A and SLE-S (Fold change = 2.0, $P < 0.05$) was defined as differentially expressed proteins (DEPs). The KEGG database and the Metascape platform were used to analyze the function and pathway enrichment of DEPs to determine the important pathways and proteins related to SLE disease activity. Protein–protein interaction (PPI) network analysis is performed by using a search tool (STRING) database for searching interacting genes/proteins.

## Qualitative identification by western blot

To verify the previous screening results, we selected C9 and MASP2 with the most obvious differences for western blot analysis. The total protein was enriched by a 10 kD ultrafiltration device (Millipore) and concentrated at 6000g for 15 min. The volume of the sample was calculated according to the protein mass (7 μg). After SDS-PAGE gel electrophoresis, the protein was transferred to the PVDF membrane and sealed with skim milk powder for 2 h. The PVDF membranes were incubated overnight at 1:1000 anti-C9 antibody and anti-MASP2 antibody at 4 °C. The 1 × TBST buffer solution was used to rinse the membranes three times, each 20 min. Then, 1:3000 s antibody was added and incubated at room temperature for 90 min. Finally, the PVDF membrane was washed with 1 × TBST buffer and detected by the enhanced chemiluminescence method.

## Quantitative identification by parallel reaction monitoring

The expression of the target proteins in urine were quantitatively determined by the parallel reaction monitoring method. Twelve urine samples were selected from each of the active group (SLE-A), the stable group (SLE-S) and the normal controls (NC), and 1 ml of urine was taken from each patient for the experiment. The urinary samples were centrifuged at 2,000 g for 10 min at 4 °C, and urea particles were added to the supernatant with a urea concentration of 8M, shaken until the urea particles were fully dissolved. The volume of the samples were concentrated by ultrafiltration in a 10 K ultrafiltration tubes. After protein quantification by the Bradford method, 60 μg proteins were taken and placed in a centrifuge tubes. A total of 5 μl of 1M DTT (5 mM/L) was added to the solution and mixed, and incubated at 37 °C for 1 h; 20 μl of 1M IAA (20 mM/L) solution was added, and after mixing, the reaction was performed for 1 h at room temperature and avoided light. Trypsin was added to the ultrafiltration tube at 1:50 enzyme-to-substrate ratio, and digested at 37 °C for more than 12 h. The mixed samples were fractionated on a Waters BEH C18 column (5 μm, 4.6 × 250 mm) by high performance liquid chromatography system (HPLC), centrifuged at 14,000 g for 20 min, and the supernatant was taken for

separation at a flow rate of 0.7 ml/min. Then, the samples were eluted with 100 µl of mobile phase A (100% ddH2O, 0.1% formic acid) and mobile phase B (100% acetonitrile, 0.1% formic acid) in a gradient dissolution. The solvent gradient was set as follows: 5% B, 0 min; 5–8% B, 5 min; 8–18% B, 30 min; 18–32% B, 27 min; 32–95% B, 6 min; 95–5%, 4 min. Finally, the eluates were monitored at UV 214 nm, collected for a tube per minute and merged into 3 fractions, which are lyophilized and waiting for Q Exactive HF-X mass spectrometer testing.

The Q Exactive HF-X mass spectrometer was operated in the data-dependent acquisition mode and there was a single full-scan mass spectrum in the Orbitrap (350–1,500 m/z, 120,000 resolution) followed by data dependent MS/MS scans in an Ion Routing Multipole at 27% normalized collision energy (HCD). The single-sample was reconstituted in 0.1% FA, and injected into the Q Exactive HF-X mass spectrometer (Thermo Fisher) operating in the Parallel reaction monitoring (PRM) mode. For PRM acquisition, primary mass spectrum (MS1) resolution was set to 120,000, and secondary mass spectrum (MS2) resolution was set to 15,000. The m/z range covered from 350 to 1,500 m/z. The maximum capacity of C-trap was set to $3 \times 10^6$, and the maximum injection was 80 ms. The dynamic exclusion range was set to 16 s and allow the mass spectrometer always operating in the parallel ion filling and detection mode. The raw data of mass spectrometry (MS) was generated and processed by Skyline software. The false discovery rate was set to 1% for proteins and peptides. The enzyme specificity was set to trypsin, and a maximum of three missed cleavages were allowed in the database search. Parameter settings for screening differential proteins: difference ratio >1.2 or difference ratio <1/1.2, $P$ value <0.05.

## Statistical analysis

Statistical analysis was performed using SPSS 21.0 software and the visualization of the data carried out by using GraphPad Prism 9.0 software. The results of quantitative data were expressed as mean ± standard deviation (SD) or median (interquartile range). The $P$-values were calculated by Student's $t$ test and Mann–Whitney $U$ test for two groups, or ANOVA (more than two groups). The correlation analysis was performed to examine the relationship between continuous variables by Spearman's correlation coefficient. Receiver operating characteristic (ROC) curve analysis and area under the curve (AUC) calculations were used to analyze the diagnostic value of urine candidate DEPs in the disease activity of patients with SLE. In all cases two-tailed $P < 0.05$ was accepted as statistically significant.

## RESULTS

### Comparison of complement-related proteins between active group (SLE-A) and stable group (SLE-S)

Screening and identifying the urine protein profile of SLE patients demonstrated that complement-related proteins showed varying degrees of difference between the active group (SLE-A) and stable group (SLE-S). To classify the functional annotations of the identified proteins, pathway analysis was performed by kyoto encyclopedia of genes and genomes (KEGG) database. The most representative pathway was complement and coagulation pathways (map04610). Among the urine protein profile, there were 16 proteins

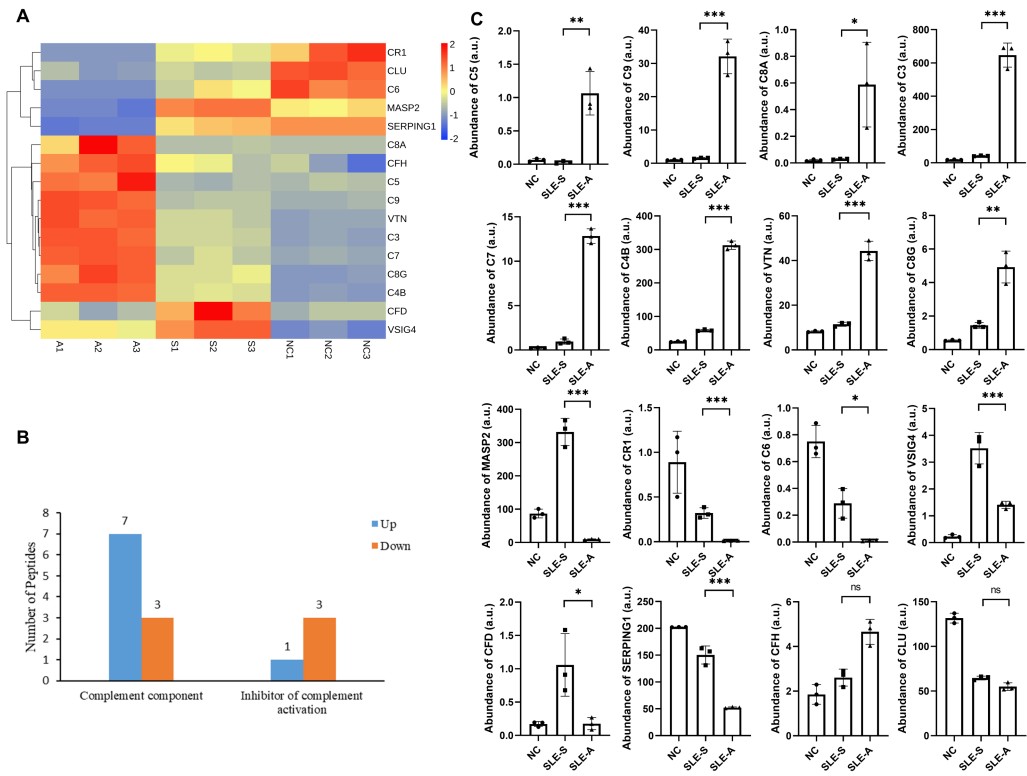

**Figure 1** **Comparison of urine proteins related to complement pathway between SLE active group and stable group.** (A) Heatmap shows the abundance peptidome related to the complement pathway among active group, stable group and normal controls. A, SLE active group; S, SLE stable group; N, normal controls; 1–3, three technical replicates for each group of mixed urine samples. (B) Classification of different roles of differential proteins in the complement pathway between active group (SLE-A) and stable group (SLE-S). The ordinate is the functional classification of complement proteins, and the abscissa is the number of proteins involved in this function. (C) The abundance of these complement related proteins among three groups in arbitrary units (a.u.). The ordinate is the group, and the abscissa is the abundance of the identified proteins. Data were shown as mean ±SEM and significance was set at $^*p < 0.05$, $^{**}p < 0.01$, $^{***}p < 0.001$, ns, not significant difference. SLE-A, SLE active group; SLE-S, SLE stable group; NC, normal controls.

involved in the complement pathway, 14 of these proteins were differentially expressed, eight differentially expressed proteins (DEPs) were up-regulated, and six DEPs were down-regulated (Fig. 1A).

These 14 DEPs played different roles in the complement pathway, 10 of these DEPs were key factors of the complement pathway. The expressions of Complement C5 (C5), Complement component C9 (C9), Complement C3 (C3), Complement component C7 (C7), Complement C4-B (C4B), Complement component C8 alpha chain (C8A) and Complement component C8 gamma chain (C8G) were up-regulated between SLE-A and SLE-S, while the expressions of Mannan-binding lectin serine protease 2 (MASP2), Complement component C6 (C6) and Complement factor D (CFD) are down-regulated. The other 4 DEPs were the inhibitory molecules of complement pathway, namely Vitronectin (VTN) whose expression was up-regulated and Complement receptor

**Table 1  The expression of complement pathway associated proteins between active group (SLE-A) and stable group (SLE-S).**

| Uinprot-ID | Protein name | Gene name | (SLE-A/SLE-S) ratio | P-value | Form of expression |
|---|---|---|---|---|---|
| P01031 | Complement C5 | C5 | 37.79 | 5.32E−03 | up |
| P02748 | Complement component C9 | C9 | 20.46 | 5.28E−04 | up |
| P07357 | Complement component C8 alpha chain | C8A | 19.94 | 3.82E−02 | up |
| P01024 | Complement C3 | C3 | 15.39 | 1.33E−04 | up |
| P10643 | Complement component C7 | C7 | 13.44 | 1.90E−05 | up |
| P0C0L5 | Complement C4-B | C4B | 5.321 | 4.00E−06 | up |
| P04004 | Vitronectin | VTN | 3.866 | 2.11E−04 | up |
| P07360 | Complement component C8 gamma chain | C8G | 3.405 | 3.38E−03 | up |
| P08603 | Complement factor H | CFH | 1.785 | 6.33E−03 | n.s. |
| P10909 | Clusterin | CLU | 0.850 | 2.51E−02 | n.s. |
| O00187 | Mannan-binding lectin serine protease 2 | MASP2 | 0.029 | 1.69E−04 | down |
| P17927 | Complement receptor type 1 | CR1 | 0.035 | 8.46E−04 | down |
| P13671 | Complement component C6 | C6 | 0.039 | 1.21E−02 | down |
| P00746 | Complement factor D | CFD | 0.167 | 3.34E−02 | down |
| P05155 | Plasma protease C1 inhibitor | SERPING1 | 0.348 | 5.62E−04 | down |
| Q9Y279 | V-set and immunoglobulin domain-containing protein 4 | VSIG4 | 0.4009 | 3.68E−03 | down |

**Notes.**

$P$ value <0.05 was considered a significant difference.

Up, Protein expression was up-regulated; Down, Protein expression was down-regulated; n.s., no significant difference.

type 1(CR1), Plasma protease C1 inhibitor (SERPING1) and V-set and immunoglobulin domain-containing protein 4(VSIG4) whose expression were down-regulated (Fig. 1B). The relative abundance of these DEPs among SLE-A, SLE-S and controls that did not have SLE (NC) were shown in Fig. 1C, and detailed information were shown in Table 1. These DEPs may play key roles in disease activity.

## Bioinformatics analysis of complement-related proteins

In order to predict the interactions of complement-related proteins, protein–protein interaction (PPI) network was conducted by STRING database. The PPI network involved in the complement pathway were tight and these proteins had strong interactions (Fig. 2A). Based on the Metascape platform, the functional analysis of these proteins revealed that the main enrichment pathways involved in complement and coagulation cascades, complement cascade, complement activation, regulation of complement cascade and other pathways. These findings suggested that these proteins play important roles in the activation and regulation of the complement pathway (Fig. 2B).

## Preliminary verification of urine C9 and MASP2 among SLE-A, SLE-S and controls group (NC)

To verify the screening results, we selected the two most different proteins from the DEPs, namely C9 and MASP2, performing western blot analysis. The results showed that the expression of C9 in the urine was significantly increased in most patients with SLE-A compared with SLE-S and NC. But the changes of MASP2 in the three groups were not obvious, and even the detection rate of MASP2 in the urine of individual SLE patients
**A**

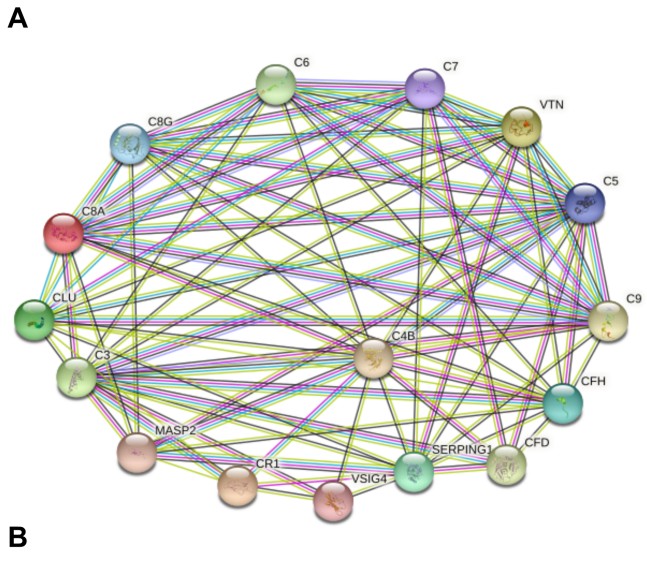

**B**

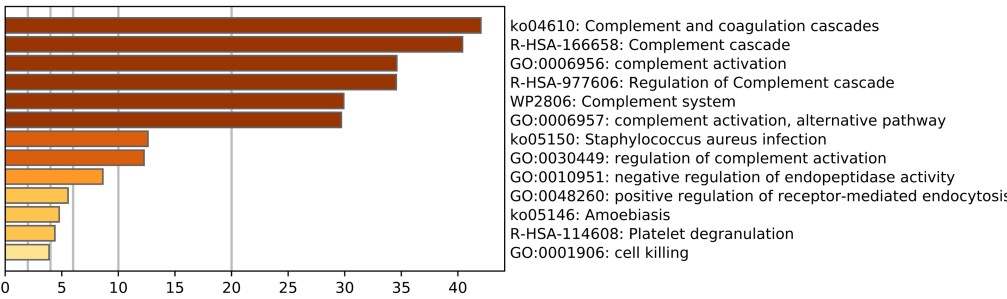

**Figure 2  Functional analysis of complement-related proteins.** (A) Protein–protein interaction(PPI) network of complement-related proteins. The confidence level was set to medium of 0.4. Lines: line color indicates the type of interaction evidence; Nodes: in the network nodes represent proteins. (B) Pathway analysis of complement-related proteins performing by Metascape platform. The vertical coordinate is the pathway code and name; -log10(P) indicates the enrichment of the pathway.

and control groups was low (Fig. S1 ). It may be due to the limitations of the western blot method in the detection of urine proteins, and the sensitivity is not enough. Therefore, we focused on the up-regulated proteins for further verification.

## Quantitative verification of up-regulated complement proteins in urine

Parallel reaction monitoring (PRM) technology was used to identified 8 up-regulated proteins related to the complement pathway. The results revealed that the expression of four proteins were consistent with the previous mass spectrometry screening results. The detailed information of the four proteins were shown in Table 2. Compared with the SLE-S and NC, the expression levels of C9, C8A, C4B, and C8G exhibited a significant increase in urine of SLE-A. The levels of these four proteins among the three groups are shown in Fig. 3A.

**Table 2** The four differentially expressed proteins (DEPs) to be verified by parallel reaction monitoring (PRM).

| Uinprot-ID | Peptide | Gene Name | M/Z | SLE-A/SLE-S | P-value | Form of expression |
|---|---|---|---|---|---|---|
| P02748 | LSPIYNLVPVK | C9 | 621.88 | 38.51 | 0.0377 | up |
| P07357 | LGSLGAACEQTQTEGAK | C8A | 860.91 | 18.80 | 0.0047 | up |
| P0C0L5 | FSDGLESNSSTQFEVK | C4B | 887.91 | 8.30 | 0.0327 | up |
| P07360 | VQEAHLTEDQIFYFPK | C8G | 655.66 | 15.83 | 0.0496 | up |

**Notes.**

Abbreviations: Up, Protein expression was up-regulated; M/Z, mass charge ratio; SLE-A, SLE active group; SLE-S, SLE stable group.

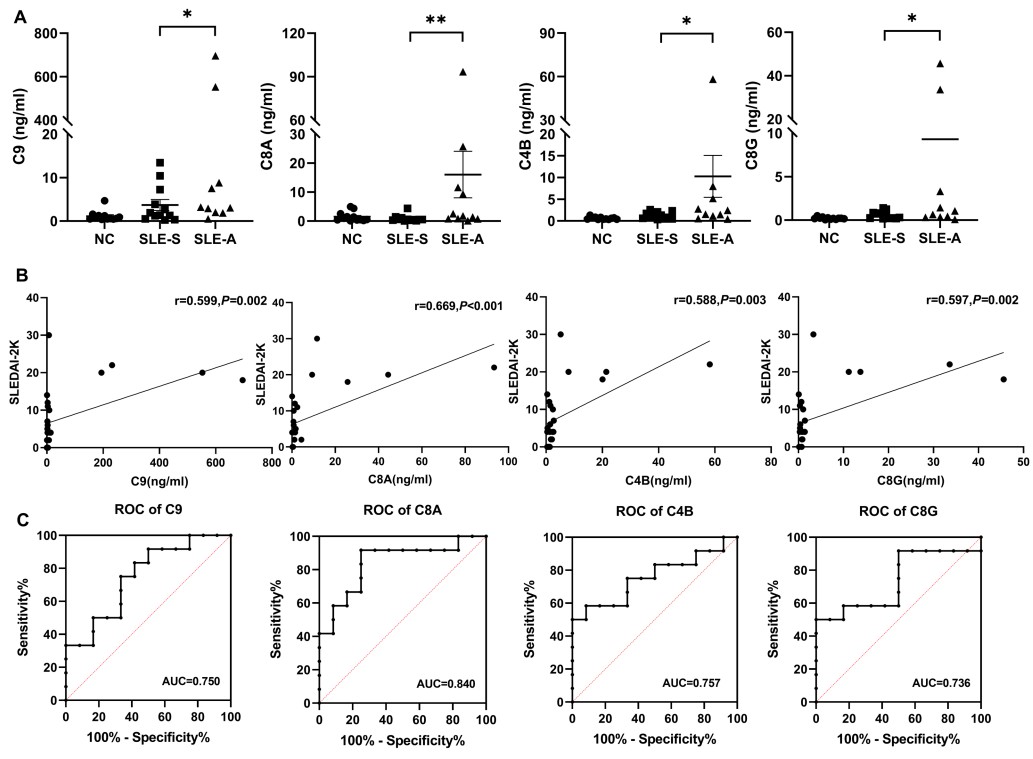

**Figure 3** The comparison of selected protein expression between SLE and normal controls (NC) by PRM. (A) The expression levels of C9, C8A, C4B and C8G among active group (SLE-A), stable group (SLE-S) and controls without SLE (NC). The ordinate is the group, and the abscissa is the detection concentration of proteins. Data were showed as mean ±SEM. An asterisk (*) indicated a significant change in protein abundance between the two groups ( $P < 0.05$ ) and two asterisk (**) meant $P < 0.01$. (B) The correlation between the signal intensity of DEPs and SLEDAI-2K, including C9, C8A, C4B and C8G, was determined by Spearman correlation analysis in all SLE patients (active group and stable group). (C) Receiver operating characteristic (ROC) curve analysis of the four DEPs to distinguish SLE active group from SLE stable group. PRM, Parallel reaction monitoring; DEPs, differentially expressed proteins; SLEDAI-2K, Systemic Lupus Erythematosus Disease Activity Index 2000; AUC, area under the curve.

## Differentially expressed proteins (DEPs) were closely associated with lupus activity

Spearman correlation analysis was used to analyze the correlation between four DEPs and disease activity. The results shown that the levels of urine C9 ($r = 0.599$, $P = 0.002$), C8A

**Table 3  Clinical performance of the four DEPs on SLE activity using SLEDAI-2K as the reference standard.**

| Index | Cut-off value (ng/ml) | Sn (%) | Sp (%) | PPV (%) | NPV (%) |
|-------|------------------------|--------|--------|---------|---------|
| C9 | 1.601 | 91.67 | 50 | 64.71 | 85.71 |
| C8A | 0.640 | 91.67 | 75 | 78.57 | 90 |
| C4B | 2.364 | 58.33 | 91.67 | 87.5 | 68.75 |
| C8G | 1.412 | 50 | 100 | 100 | 66.67 |

**Notes.**

Abbreviations: Sn, Sensitivity; Sp, Specificity; PPV, Positive predictive value; NPV, Negative predictive value.

$(r = 0.669, P < 0.001)$, C4B $(r = 0.588, P = 0.003)$, and C8G $(r = 0.597, P = 0.002)$ were significantly positively correlated with SLEDAI-2K. The higher the score of SLEDAI-2K, the greater the concentration of these four proteins (Fig. 3B).

ROC curve analysis was performed to determine the cutoff value of the four DEPs for distinguishing SLE-A from SLE-B. The areas under the curves(AUCs) of urine C9, C8A, C4B and C8G were 0.750 (95% CI [0.554–0.947]), 0.840 (95% CI [0.674–1.00]), 0.757(95% CI [0.556–0.958]) and 0.736 (95% CI [0.526–0.946]), respectively (Fig. 3C). These four proteins all have high discriminatory capacity for disease activity. We evaluated the clinical performance of these four DEPs for SLE disease activity, including sensitivity, specificity, positive predictive value and negative predictive value, using SLEDAI-2K as the reference standard, as shown in Table 3.

Among the four urinary proteins, C8A had the highest judgment value for lupus activity, with an AUCs of 0.840. When the cutoff level was 0.640 ng/ml, the sensitivity, specificity, positive predictive value and negative predictive value were 91.67%, 75%, 78.57% and 90%, respectively.

# DISCUSSION

Although the pathogenesis of SLE is not fully understood, it is widely accepted that the complement system in the blood is involved in the pathogenesis of SLE and has important roles in the pathophysiology of SLE, including promotion of inflammatory processes, clearance of immune complexes, cellular and apoptotic debris (*Walport, 2001a*; *Weinstein, Alexander & Zack, 2021*; *Cook & Botto, 2006*).The complement system is composed of more than 30 plasma proteins and cell surface receptors. There are three pathways of complement activation: classical pathway, alternative pathway and mannose-binding lectin pathways, which may be related to inflammation and tissue damage in SLE (*Walport, 2001b*; *Kim et al., 2020*; *Troldborg et al., 2018*). Therefore, activating components or products of complement in SLE patients may serve as biomarkers for diagnosis or monitoring disease activity of SLE.

In the past few decades, most studies have focused on the detection of complement proteins (C3, C4) and hemolytic activity. Although known as markers of disease activity, the limitations of these indicators have also caused some controversy (*Schur & Sandson, 1968*; *Esdaile et al., 1996a*; *Esdaile et al., 1996b*). In recent years, some studies have found that detection of complement split products more accurately reflected complement activation,

such as C3a, C4a, C5a, C3d, C4d, Ba, Bb and soluble C5b-9 complexes (*Morell, Perez-Cozar & Maranon, 2021*; *Liu, Ahearn & Manzi (2004)*; *Porcel et al., 1995*). However, these split products have a short half-life in plasma and are easily activated *in vitro*, so there are many challenges in detection technology (*Liu et al., 2004*). The content of proteins in urine is relatively stable, and urine is easy to collect and available in large quantities (*Decramer et al., 2008*). Therefore, the detection of complement components and split products in urine is a convenient and good choice.

Several investigators reported that the complement split products like C3 fragment, C4d, C5a and C5b-9 could be detected in the urine of lupus nephritis (LN) patients, and found that the levels of C3d in urine was closely related to LN disease activity (*Mejia-Vilet et al., 2021*; *Tamano et al., 2002*). However, the changes of other complement pathway-related proteins and fragments in the urine of SLE patients are unknown. Therefore, in order to explore the changes of complement components in the urine of SLE patients and the relationship with SLE disease activity, and search the potential urine markers of monitoring SLE activity, we analyzed the urine protein profiles of patients with active lupus and stable lupus, and screened out the urinary proteins involved in the complement pathway, based on label-free mass spectrometry technology. Subsequently, we further investigated the expression levels and functions of the complement related DEPs between the active group (SLE-A) and stable group (SLE-S). Then, the up-regulated DEPs were selected for quantitative verification using targeted proteomics technology to explore the clinical application value of these urinary proteins as biomarkers of disease activity.

In this study, a total of 16 complement pathway-related proteins were identified by analyzing the urine protein profiles of SLE, 14 of these proteins were abnormally expressed in the active and stable SLE groups. Through functional analysis, we found that these differential proteins have different roles in the complement pathway, mainly involved in complement activation and the regulation of complement activation pathway, and these DEPs may as crucial markers for SLE disease activity. To verify the expression of these DEPs in the urine of SLE patients, we focused on the verification of 7 up-regulated proteins relating to complement activation pathway (C5, C9, C8A, C3, C7, C4B and C8G). Our results showed that the expression of urinary C9, C8A, C4B and C8G in SLE patients was consistent with previous mass spectrometry data.

C9 and C8 (C8A, C8G) are both the final components of the complement system, major components of the membrane attack complex (MAC), which served as part of innate and adaptive immune responses (*Dudkina et al., 2016*). C8 contains three polypeptides, ($\alpha$, $\beta$, and $\gamma$), consisting of disulfide-linked heterodimers ($\alpha$- $\gamma$) and non-covalently bound $\beta$ chains (*Hadders, Beringer & Gros, 2007*). C9 binds C8 in the C5b-8 complex and forms the MAC that causes cellular and organ damage, therefore, the complement system is involved in the occurrence and development of various diseases (*Zhao et al., 2021*). Recent evidence reported that the decrease of eGFR in patients with diabetic nephropathy was significantly associated with the excretion of the urinary membrane attack complex (*Pelletier et al., 2019*). Our results revealed that urinary complement C9, C8A and C8G were significantly increased during active SLE, and correlated well with SLEDAI-2K, suggesting complement activation and the formation of membrane attack complexes during SLE disease activity.
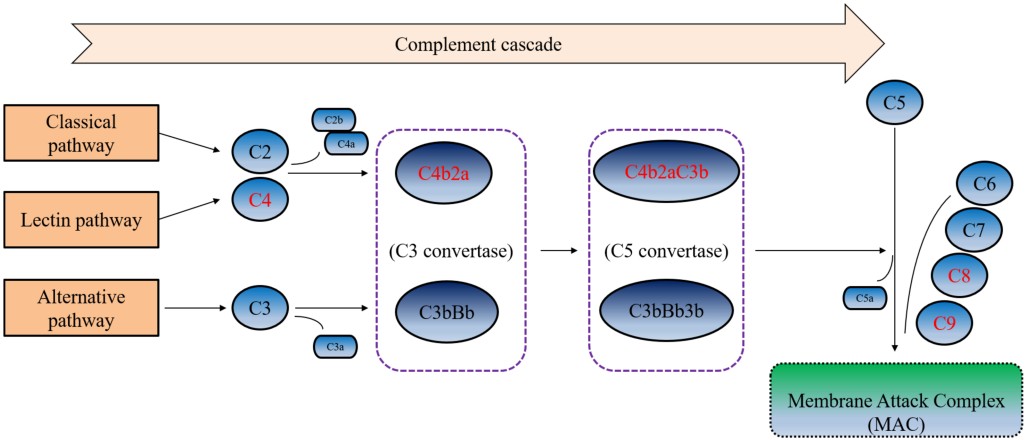

**Figure 4  Simplified complement cascade featuring complement-related proteins.** C9, C8A, C4B and C8G play different roles in the complement pathway. Red indicating the four significantly up-regulated proteins or protein complex.

C4 is a non-enzymatic component of C3 and C5 convertases and is necessary for complement activation (*Pelletier et al., 2019*). After complement was activated through the classical or lectin-activated pathway, C4 was activated and hydrolyzed to generate C4B. C4B was an opsonin that covalently binds complement-activating targets (*Mayilyan et al., 2008*). Recent studies have reported that diversities of C4A and C4B proteins and their gene copy number variations (CNVs) in healthy subjects and patients with autoimmune diseases (*Zhou et al., 2021*). Our results demonstrated that the expression of C4B was also significantly elevated in the urine of active SLE, and was positively correlated with disease activity.

Furthermore, in the ROC curve analysis of urine C9, C8A, C8G and C4B on SLE disease activity, we found that these four proteins all have good ability to judge SLE activity. Among these proteins, urine C8A had the best judgment ability, with an AUC of 0.840. When the cut-off value was 0.640 ng/ml, the sensitivity and specificity for judging disease activity were 91.67% and was 75%, respectively. These four complement related DEPs undertook critical roles and performed different functions in the complement cascade (Fig. 4). As the important component of C3 convertase and C5 convertase, C4b played a very important role in the entire complement cascade. While C9, C8A, and C8G were mainly involved in the last step of the complement cascade, which participated in the formation of membrane attack complexes, resulting in cell lysis and tissue damage. According to our results, urine C9, C8A, C8G, and C4B were significantly increased during active SLE, the four complement-related proteins may be potential urinary biomarkers of SLE disease activity.

There are certain limitations of this study. First, the small sample size enrolled in this study does not allow for correction of confounding factors. Further expansion of the sample size is needed to analyze clinical values and confounding factors in the future. Second, the methodological limitations of mass spectrometry, when we use this technique to analyze and

validate urinary proteins, we cannot detect peptides with too high or too low abundance, which may eventually lead to the loss of information of some important proteins. Third, this study is a cross-sectional study and lacks the validation of a longitudinal research. Therefore, in the follow-up study, we will try our best to expand the sample size and continue to focus on the changes and expression of these complement factors in the urine of SLE patients, and analyze the clinical application value of these potential biomarkers from multiple perspectives using a multi-omics approach such as transcriptome and genome, *etc*.

## CONCLUSIONS

In conclusion, we analyzed and verified the expression of complement pathway-related proteins in the urine of SLE patients with different activity states by using proteomic technology. Then, we further explored the association between these complement-related proteins and disease activity. Higher abundance of urinary C9, C8A, C8G, and C4B were strongly associated with higher scores of SLEDAI-2K. These findings suggested that activation of distinct components of the complement pathway may be associated with disease activity. Consequently, therapeutically targeting the complement pathway may improve progression of SLE and these four urinary proteins may be potential biomarkers to assist in monitoring lupus activity.

### Statement of Ethics

This study was approved by the ethics committee of Beijing Shijitan Hospital, Capital Medical University. The participants all gave written informed consent of each subject, which was in accordance with the provisions of the Helsinki Declaration.

### Funding

This work was supported by the Beijing Key Clinical Specialty Program (grant number: 2020ZDZK2). The funders had no role in study design, data collection and analysis, decision to publish, or preparation of the manuscript.

### Grant Disclosures

The following grant information was disclosed by the authors:
Beijing Key Clinical Specialty Program: 2020ZDZK2.

### Competing Interests

The authors declare there are no competing interests.

### Author Contributions

- Jin Zhao conceived and designed the experiments, performed the experiments, prepared figures and/or tables, authored or reviewed drafts of the article, and approved the final draft.

- Jun Jiang conceived and designed the experiments, performed the experiments, prepared figures and/or tables, authored or reviewed drafts of the article, and approved the final draft.
- Yuhua Wang analyzed the data, authored or reviewed drafts of the article, and approved the final draft.
- Dan Liu analyzed the data, prepared figures and/or tables, and approved the final draft.
- Tao Li analyzed the data, prepared figures and/or tables, and approved the final draft.
- Man Zhang conceived and designed the experiments, prepared figures and/or tables, authored or reviewed drafts of the article, and approved the final draft.

### Human Ethics

The following information was supplied relating to ethical approvals (*i.e.*, approving body and any reference numbers):

This study protocol was approved by the ethics committee of Beijing Shijitan Hospital, Capital Medical University (Ethical Application Ref: sjtkyll-lx-2021(55)).

### Data Availability

The raw data of mass spectrometry are available in the Supplementary Files.

### Supplemental Information

Supplemental information for this article can be found online at http://dx.doi.org/10.7717/peerj.14383#supplemental-information.

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
