# Peer review of "Significance of urine complement proteins in monitoring lupus activity"

_PeerJ, doi:10.7717/peerj.14383_

## Round 0.1 · original submission · Minor Revisions

Please follow the suggestions of the reviewers and resubmit your manuscript at the earliest. Looking forward to the revised version.

Reviewer 1 ·

Basic reporting

no comment

Experimental design

The authors need to clarify some of the points as follows:
Line 110: How much volume of urinary samples was used per patient?
Line 113: What method was used to quantify protein?
Line 114-115: What is the final concentration of DTT and IAA used?
Line 119: What column was used for separation of the mix-sample?

Validity of the findings

no comments

Additional comments

The study is very interesting. It is well rounded but needs few method details for reproducibility.

Reviewer 2 ·

Basic reporting

In this study, authors have evaluated Complement system proteins in the urine samples of SLE patients, have looked at differentially expressed proteins in active phase of the disease and have also explored their clinical relevance.

Experimental design

OK

Validity of the findings

OK

Additional comments

1. Sample size looks small to make conclusions about clinical relevance.
2. Materials and methods could have been better presented. The legends for the figures are poorly written. It is extremely difficult to figure out what has been plotted on X and Y axis and for which group.
3. In fig1c, the data points are just three in numbers. What are these three points; replicates,/ no. of subjects or any other parameter?
4. Fig3 legends are similarly ambiguous (3b, 3C). Data from which group has been plotted in 3b and 3C is not very clear. I would request authors to tabulate the % Sensitivity and specificity for Fig3C.
5. Authors should also present Positive and Negative predictive value (PPV and NPV) for determining clinical relevance.
6. What were the confounding factors and the limitations of the study should be discussed in detail.
7. Urine analysis in SLE patients basically predicts nephritis. What was the status of Complement proteins expression in the patients? A bioinformatic analysis with publicly available database for single cell RNA-data seq will help provide better picture as predictor of nephritis.

---

## Round 0.2 · accepted · Accept

As the authors gave a minor revision decision on this manuscript and the authors have incorporated the revisions I conclude the revisions are satisfactory and consider the manuscript for publication.